# The Effect of the Bending Beam Width Variations on the Discrepancy of the Resulting Quadrature Errors in MEMS Gyroscopes

**DOI:** 10.3390/mi13050655

**Published:** 2022-04-20

**Authors:** Alexandre Azier, Najib Kacem, Bernard Chaumet, Noureddine Bouhaddi

**Affiliations:** 1Department of Applied Mechanics, FEMTO-ST Institute, CNRS/UFC/ENSMM/UTBM, University Bourgogne Franche-Comté, 25000 Besançon, France; alexandre.azier@fr.thalesgroup.com (A.A.); noureddine.bouhaddi@univ-fcomte.fr (N.B.); 2Thales Avionics, 26027 Valence, France; bernardchaumet5@gmail.com

**Keywords:** MEMS, gyroscope, inertial sensor, quadrature signal, technology defects

## Abstract

In this paper, we develop a new approach in order to understand the origin of the quadrature error in MEMS gyroscopes. As the width of the flexure springs is a critical parameter in the MEMS design, it is necessary to investigate the impact of the width variations on the stiffness coupling, which can generate a quadrature signal. To do so, we developed a method to determine the evolution of the stiffness matrix of the gyroscope springs with respect to the variation of the bending beams width of the springs through finite element analysis (FEA). Then, a statistical analysis permits the computation of the first two statistical moments of the quadrature error for a given beam width defect. It turns out that even small silicon etching defects can generate high quadrature level with up to a root mean square (RMS) value of 1220°/s for a bending beam width defect of 0.9%. Moreover, the quadrature error obtained through simulations has the same order of magnitude as the ones measured on the gyroscopes. This result constitutes a great help for designing MEMS gyroscopes, as the consideration of the bending beams width defects is needed in order to avoid high quadrature error.

## 1. Introduction

Silicon Micro-electromechanical systems (MEMS) are very attractive for many uses due to their miniaturization and their performances. Although these devices are already present in many everyday objects, such as smartphones, their use for navigation applications in space, aeronautics, or autonomous vehicles remains a real scientific and technological challenge. Indeed, the technological advances made on MEMS have effectively made it possible to approach, although still without achieving, the levels of precision sought for this type of application. As a matter of fact, in the world of high performance, MEMS sensors must be 10,000 times more accurate than consumer sensors. Thereby, it is very challenging to design and manufacture silicon MEMS gyroscopes with performance levels of tactical and navigation grades [1,2], which are so far reserved to ring laser gyroscopes (RLGs) [3,4], fiber optic gyroscopes (FOGs) [4,5,6] and hemispherical ring gyroscopes (HRGs) [7,8].

Since the introduction of first MEMS rate gyroscopes and accelerometers thanks to the work of miniaturization of inertial systems carried out by Draper Laboratory in the late 1980s [9,10], the precision of MEMS gyroscopes has improved a lot in the last decades, from a bias of few hundred degrees per hour [9,11] to few degrees per hour [12,13,14,15]. Moreover, as a result of all of these years of research in MEMS gyroscope mechanical design and to the unceasing improvements in microfabrication, silicon, high-quality packaging, and electronics technology, MEMS gyroscopes with 0.1°/h bias and $0.01{}^{\circ}/\sqrt{h}$ ARW (angular random walk) are now a reality [16,17,18,19].

Vibrating gyroscopes consist of one or more mobile vibrating masses [17,20,21] connected to each other and to their support by bending beams (which act as springs) in order to constitute an excitation resonator and a detection resonator, the two being coupled to each other by the Coriolis acceleration [22]. Thus, when the gyroscope rotates around its sensitive axis, the composition of the forced vibration with the angular rotation vector generates forces, through the Coriolis effect, which induces vibration of the moving masses in the axis orthogonal to sensitive axis and forced vibration axis. This vibration is then detected by a detection transducer, the electrical signals of which are exploited by an electronic circuit to deduce therefrom a value of the angular speed around the sensitive axis. Therefore, any parasitic force exerted on the resonator with the same phase and frequency as the Coriolis force and in the same direction as the Coriolis induced vibration will results, if it is not compensated, in an error on the sensor output called bias or zero-rate output (ZRO) [23]. Even though the bias can be removed from the final output signal with calibration, it varies over time and is sensitive to temperature variations and external vibrations (this phenomenon is called bias instability). So, in practical cases, its absolute value is minimized in order to reduce its instability. To do so, it is necessary to identify the factors that contribute to the bias of the gyroscope.

One of the major error sources is the mechanical quadrature signal [23,24,25,26]. The quadrature is an unwanted force resulting from the coupling stiffness between the excitation-mode displacement and the sensitive-mode of the gyroscope. This force has the same frequency as the Coriolis signal, but is in phase quadrature (90° phase shift) with it, hence its name [24,25,26,27]. Although its amplitude can be far more important than the amplitude of the Coriolis signal, it is possible to set its impact on the output signal aside. Indeed, one can use the 90° phase shift between the Coriolis and the quadrature signal by using a phase sensitive demodulation via an electronic processing. By doing so, it is possible to split the sense information in two parts, so we can retrieve the Coriolis output [28]. However, even a slight phase error on the demodulation $\left(<1{}^{\circ}\right)$ can generate an important bias [29,30] and the high amplitude of the Coriolis signal can saturate the input accepted by the demodulator [31]. It is then necessary to reduce the amplitude of the quadrature signal.

One way to do so is to cancel the quadrature motion of the resonator using sense feedback electrodes. This can be carried out by applying an electrostatic force to the sense combs, with the same amplitude and in phase opposition with the quadrature force [28,32]. Else, the quadrature signal can be compensated before the sense signal being demodulated. One can do so by adding a compensation signal with the same amplitude and in phase opposition with the quadrature force via a dedicated close loop [33]. Both of these methods work properly, but their feedback needs to be modulated, still letting us with the phase error problem. To overcome this issue, dedicated correction combs can be used to cancel the mechanical coupling stiffness. The idea is to apply the right DC voltage on the correction combs so the electrostatic stiffness of these combs has the exact opposite value of the mechanical coupling stiffness [27,34,35,36].

Whatever the compensation method(s) chosen, it is important to design a well-balanced resonator, while taking into account the manufacturing defects. Indeed, on most of the MEMS gyroscope designs, one or several springs are place symmetrically on each corner of the resonator so that if all of the springs are identical, the resulting coupling stiffness is null [30]. However, if there is geometrical dispersion of the flexure springs, especially width variation as it is a critical dimension for the springs, the resulting coupling stiffness would not be equal to zero [37], which would generate a quadrature force. To our knowledge, few investigations have been carried out about the contributions of the geometrical dispersion of the springs on quadrature error. For instance, in [27], MEMS gyroscopes have been designed with intentional spring imbalances. It emerges that the measure of the quadrature outputs of the fabricated gyroscopes, are in good agreement with FEA. Also, the relative intrinsic coupling stiffness $K_{xy}/K_{x}$ of different types of springs has been evaluated, with and without a width reduction, by finite element simulations in [30]. It seems that the quadrature error generated by a local width defects can be reduced by choosing specifics spring designs.

Still, on a real gyroscope, due to the manufacturing defects, each spring has its own width, so the designs simulated and/or fabricated in [27,30] are far from reflecting the reality. A more relevant study has been brought in [38], where Monte Carlo Simulation (MCS) was performed, in which the width of the eight springs and the gaps in the four combs of the considered design were assumed to be independent and normally distributed random variables. The results showed that variations of the beam width lead to a significant discrepancy between the resulting quadrature errors. Nevertheless, this result is mixed by the fact that the main contribution to the quadrature error is the asymmetric topology of half of the springs of this particular gyroscope, so the width variation on this half is more prone to induce quadrature error than the width variation of the other half. Besides this issue, the whole model on the gyroscope was simulated for each sample of the MCS which is time consuming, thereby limiting the number of simulations.

In this paper, we propose a method to evaluate the impact of silicon etching defects on the amplitude of the quadrature signal. To do so, we perform simulations using a finite element model (FEM) in order to compute the stiffness matrix of the springs of the gyroscope, regarding the bending beams width variations. Then, MCS was performed in order to investigate the impact of the spring width variation on the discrepancy between the resulting quadrature errors with a reduced computational time. This is achieved by reducing the complexity of the simulations and by using simple analytic formulas. These results are used to calculate the first two statistical moments of the quadrature error, which are compared to the experimental results, obtained using a dedicated bench.

## 2. Gyroscope Dynamics

The dynamic equations of a linear vibrating gyroscope can be expressed as [24,39]:(1)$$\left[\begin{array}{cc} M & 0\\ 0 & M \end{array}\right]\frac{d^{2}\vec{U}}{dt^{2}}+\left(\left[D\right]+\left[C\right]\right)\frac{d\vec{U}}{dt}+\left[\begin{array}{cc} K_{x} & K_{xy}\\ K_{xy} & K_{y} \end{array}\right]\vec{U}=\vec{F}$$
where M is the modal mass, $\vec{U}=\left(\begin{array}{c} x\\ y \end{array}\right)$ represents the displacement vector, $\left[D\right]=\left[\begin{array}{cc} D_{x} & D_{xy}\\ D_{yx} & D_{y} \end{array}\right]$ represents the damping matrix, $\left[C\right]=\left[\begin{array}{cc} 0 & -2M\Omega_{z}\\ 2M\Omega_{z} & 0 \end{array}\right]$ represents the Coriolis matrix, $\Omega_{z}$ is the angular velocity around the *z*-axis, $K_{x}$ and $K_{y}$ represents respectively the stiffness along the *x* and *y*-axis, $K_{xy}$ the coupling stiffness term bringing the quadrature signal and $\vec{F}=\left(\begin{array}{c} F_{x}\\ F_{y} \end{array}\right)$ the driving vector force. In the *y*-axis, the Equation (1) leads to:(2)$$M\ddot{y}+D_{y}\dot{y}+K_{y}y=-K_{xy}x-K_{yx}x-2M\Omega_{z}\dot{x}+F_{y}$$

It is then necessary to understand the origins and to calculate the $K_{xy}$ term in order to deduce the amplitude of the quadrature signal. In this paper, we propose to evaluate the impact of silicon etching imperfections on the coupling stiffness term.

## 3. Expression of the Coupling Stiffness

For confidentiality purposes, the only representation of our gyroscope design we can give is the one show in Figure 1; the numbered springs $S{\left(M1\right)}^{i}$ (respectively, $S{\left(M2\right)}^{i}$ and $S{\left(C\right)}^{i}$) are fixed to the anchor points A (respectively, to the anchor points A and to the M2 mass) to one end and tied to the M1 mass (respectively, to the M2 mass and to the M1 mass) to the other. In the drive (respectively, sense) mode, the masses M1 and M2 vibrate out of phase along the x-axis (respectively, y-axis).

As we focus on the mechanical stiffness of the resonator, electrostatic combs are not taken into account in our study (their mass, though, are include in the total mass of the structure). In order to describe the stiffness matrix of the flexure springs, two coordinate systems are defined as follows:
$B\left(\vec{x};\vec{y}\right)$ is the canonical coordinate system (the *x* and *y*-axis);$B{’}_{i}\left(\vec{x_{i}};\vec{y_{i}}\right)$ represents the eigenbase of the spring number *i*.

Thus, the transition matrix $P_{i}$ between B and $B{’}_{i}$ is expressed by:(3)$$P_{i}=\left[\begin{array}{cc} \cos\left(\theta_{i}\right) & -\sin\left(\theta_{i}\right)\\ \sin\left(\theta_{i}\right) & \cos\left(\theta_{i}\right) \end{array}\right]$$

Moreover, we define ${\left[k\right]}_{B}^{i}$ and ${\left[k\right]}_{B{’}_{i}}^{i}$ the stiffness matrix of the *i*th spring wrote in the coordinate system B and $B{’}_{i}$ respectively. Then, we have:(4)$${\left[k\right]}_{B}^{i}=P_{i}{\left[k\right]}_{B’}^{i}P_{i}{}^{T}$$

By substituting the Equation (3) into the Equation (4) we can deduct the total stiffness matrix ${\left[K\left(M1\right)\right]}_{B}$ of the vibrating mass M1, described in the B coordinate system:(5)$${\left[K\left(M1\right)\right]}_{B}=\left[\begin{array}{cc} K{\left(M1\right)}_{x} & K{\left(M1\right)}_{xy}\\ K{\left(M1\right)}_{xy} & K{\left(M1\right)}_{y} \end{array}\right]=\sum_{i=1}^{4n}{\left[k\left(M1\right)\right]}_{B}^{i}=\sum_{i=1}^{4n}\left[\begin{array}{cc} k{\left(M1\right)}_{x}^{i} & k{\left(M1\right)}_{xy}^{i}\\ k{\left(M1\right)}_{xy}^{i} & k{\left(M1\right)}_{y}^{i} \end{array}\right]$$
with:(6)$$K{\left(M1\right)}_{x}=\sum_{i=1}^{4n}k{\left(M1\right)}_{x}^{i}=\sum_{i=1}^{4n}(a{\left(M1\right)}_{i}+{\left(-1\right)}^{i}b{\left(M1\right)}_{i}\cos\left(2\theta_{i}\right))\;K{\left(M1\right)}_{xy}=\sum_{i=1}^{4n}k{\left(M1\right)}_{xy}^{i}=\sum_{i=1}^{4n}{\left(-1\right)}^{i}b{\left(M1\right)}_{i}\sin\left(2\theta_{i}\right)\;K{\left(M1\right)}_{y}=\sum_{i=1}^{4n}k{\left(M1\right)}_{y}^{i}=\sum_{i=1}^{4n}(a{\left(M1\right)}_{i}-{\left(-1\right)}^{i}b{\left(M1\right)}_{i}\cos\left(2\theta_{i}\right))\;a{\left(M1\right)}_{i}=\frac{k{\left(M1\right)}_{I}^{i}+k{\left(M1\right)}_{II}^{i}}{2}\;b{\left(M1\right)}_{i}=\frac{k{\left(M1\right)}_{I}^{i}-k{\left(M1\right)}_{II}^{i}}{2}$$
where $k{\left(M1\right)}_{I}^{i}$ and $k{\left(M1\right)}_{II}^{i}$ being the eigenvalues of the stiffness matrix of the *i*th spring of M1, i.e., the spring $S{\left(M1\right)}^{i}$ to use the same notation as in Figure 1, and 4n the total number of flexure springs attached to M1. As the total stiffness matrix ${\left[K\left(M2\right)\right]}_{B}$ of the vibrating mass M2 and the total stiffness matrix ${\left[K\left(C\right)\right]}_{B}$ of the coupling springs can be calculated by using similar equations as Equations (5) and (6), it is then possible to evaluate the terms $K_{x}$, $K_{y}$ and $K_{xy}$ of the matrix ${\left[K\right]}_{B}$. This represents the stiffness matrix of the operating mode of our gyroscope, i.e., the anti-phase mode [16].

Therefore, in order to calculate the value of the coupling stiffness, it is first necessary to determine the evolution of $k{\left(M1\right)}_{x}^{i}$, $k{\left(M1\right)}_{y}^{i}$ and $k{\left(M1\right)}_{xy}^{i}$ with respect to the variation of the silicon etching imperfections, which can be carried out through FEM simulations.

A numerical three-dimensional finite element model has been developed on ANSYS^TM^ V 19.2 to simulate the static mechanical response of the flexure springs. Our model consists of four Silicon springs, clamped on one end and subjected to a static load on the other end via a rigid remote point, as represented in Figure 2. The width dimensions, illustrated in Figure 3, of each springs can be changed in a different manner for each springs, but the length l and the thickness h of the beams remain the same for all beams.

This model was meshed using eight-node elements (HEX8). In total, up to 212,000 elements and 276,000 nodes were used. It should be noted that before carrying out simulations, a mesh sensitivity study was performed to ensure the convergence of the finite element simulation results. Degrees of freedom of the model’s nodes are 3D displacements x, y and z. As we are keen to know the evolution of the coupling stiffness $k{\left(M1\right)}_{xy}^{i}$ with respect to $e_{x}$ and $e_{y}$, two steps of simulations are required. Figure 4 illustrates the simulation procedure.

For the first step, only the flexure springs 1 and 3 of the Figure 2 are simulated and their widths are equal, i.e., $e_{x}=e_{y}=e_{0}$. In such geometrical configuration, springs 1 and 3 behave as pure linear spring when a load is applied along the $X_{1}$ and $Y_{1}$ axis, also respectively called the T and N axis. These axes correspond to the diagonals of the square shaped by the four beams of a flexure spring. Then, we can run two simulations, which simply consist of applying a load F on the remote point linked to the spring 1 and 3 along the T (respectively, N) axis and retrieving the displacement $d_{T}$ (respectively, $d_{N}$) of the remote point along the same axis. So we have:(7)$$k{\left(M1\right)}_{I}^{1}=k{\left(M1\right)}_{I}^{3}=\frac{F}{2d_{T}}\;k{\left(M1\right)}_{II}^{1}=k{\left(M1\right)}_{II}^{3}=\frac{F}{2d_{N}}$$

For the second step, all flexure springs represented in Figure 2 are simulated, the widths of the spring 2, 3 and 4 are equal to $e_{0}$ (the same $e_{0}$ as in the first step) and the widths $e_{x}$ and $e_{y}$ of the spring 1 are not identical, i.e., $e_{x}\neq e_{y}$, but have a value close to $e_{0}$. Thereafter, we run two simulations, in which we set up a load F on the remote point connected to all of the springs along the x (respectively, y) axis and retrieving the displacement $\left(X_{1};Y_{1}\right)$ (respectively, $\left(X_{2};Y_{2}\right)$) of the remote point along the x and y axes. Next, let $\left[R\right]$ denotes the stiffness matrix of our four springs-remote point system:(8)$$\left[R\right]=\left[\begin{array}{cc} R_{x} & R_{xy}\\ R_{xy} & R_{y} \end{array}\right]=\sum_{i=1}^{4}{\left[k\left(M1\right)\right]}_{B}^{i}$$

Thanks to the results of the previous simulations, we are able to calculate the terms of the matrix $\left[R\right]$:(9)$$R_{x}=\frac{F}{X_{1}}\left(1+\frac{Y_{1}{}^{2}}{\left(Y_{2}X_{1}-X_{2}Y_{1}\right)}\right)\;R_{xy}=\frac{-FY_{1}}{\left(Y_{2}X_{1}-X_{2}Y_{1}\right)}\;R_{y}=\frac{FX_{1}}{\left(Y_{2}X_{1}-X_{2}Y_{1}\right)}$$

Using the values of $k{\left(M1\right)}_{I}^{i}$ and $k{\left(M1\right)}_{II}^{i}$ obtained in the first step, for i=2, 3 or 4, we can easily compute the values of $k{\left(M1\right)}_{x}^{i}$, $k{\left(M1\right)}_{xy}^{i}$ and $k{\left(M1\right)}_{y}^{i}$ thanks to Equations (3)–(6) and (7). So, we are now able to calculate the value of ${\left[k\right]}_{B}^{1}$, the stiffness matrix of the spring 1, as we know the value of each term of $\left[R\right]$ and of ${\left[k\right]}_{B}^{i}$ for the springs 2, 3 and 4:(10)$${\left[k\left(M1\right)\right]}_{B}^{1}=\left[R\right]-\sum_{i=2}^{4}{\left[k\left(M1\right)\right]}_{B}^{i}$$

Then, we can run more simulations varying the value of $e_{x}$ and $e_{y}$, ($e_{0}$ could also be changed if needed) while remaining in the permissible range of the silicon etching defects, in order to construct a database with the values of ${\left[k\left(M1\right)\right]}_{B}^{1}$ terms. Once we have arried out a certain number of finite element analyses by varying the widths the spring 1, we can determine the evolution of $k{\left(M1\right)}_{x}^{1}$, $k{\left(M1\right)}_{y}^{1}$ and $k{\left(M1\right)}_{xy}^{1}$ as a function of ε, the value of the implemented width defects ($e_{x}=e_{0}-\varepsilon$ and $e_{y}=e_{0}+\varepsilon$), by fitting polynomials (in ε) to the constructed database.

The results are shown in Figure 5, (the nominal values $e_{0}$, $k_{x}^{0}$, $k_{xy}^{0}$ and $k_{y}^{0}$ cannot be provided for confidentiality reasons). We also want to place emphasis on the fact that the evolution of the $k{\left(M1\right)}_{x}$ and $k{\left(M1\right)}_{y}$ terms as a function of ε is the same for the spring 1 to 4, only the evolution of the $k{\left(M1\right)}_{xy}$ is different for the even-numbered and the odd-numbered spring $\left(k{\left(M1\right)}_{xy}^{1}=k{\left(M1\right)}_{xy}^{3}=-k{\left(M1\right)}_{xy}^{2}=-k{\left(M1\right)}_{xy}^{4}\right)$. Also, each terms of ${\left[K\left(M2\right)\right]}_{B}^{i}$ and ${\left[K\left(C\right)\right]}_{B}^{i}$ can be determined by applying the same procedure.

As long as the value of the stiffness matrix terms of springs 2, 3, and 4 are known, the proposed simulations could be applied to any kind of beams. Then, the stiffness matrix of spring 1 can be deduced for an implemented width defect. Otherwise, the first step could be used to characterize the stiffness matrix terms of springs 2, 3, and 4.

## 4. Impact on the Amplitude of the Quadrature Signal

Knowing the evolution of the stiffness matrix terms of each spring versus the width variation of the bending beams, we can perform statistical calculations. Indeed, the manufacturing process being not perfect, a different error is made of the width on each beam of each springs of the gyroscope. Here, we assume that this error follows a normal law $N\left(0;\varepsilon /3\right)$ (ε/3 is as qualitative value, so that 99.7% of the bending beams have a width comprise between $\left(e_{0}-\varepsilon \right)$ and $\left(e_{0}+\varepsilon \right)$) for $e_{x}$ and $e_{y}$ of each spring. Therefore, based on Equation (5) and the evolution formulas of the stiffness matrix terms of each spring versus the width variation, $10^{5}$ samples (a lot of samples are used in order to obtain accurate statistical values) are carried out via a custom-built Matlab^TM^ program. Then, the mean and the standard deviation of the normalized amplitude of the quadrature signal $K0_{Q}$ caused by the coupling stiffness are calculated (with $\omega_{0}{}^{2}=\frac{\omega_{x}{}^{2}+\omega_{y}{}^{2}}{2}$, $\omega_{x}=2\pi \sqrt{\frac{K_{x}}{M}}$ and $\omega_{y}=2\pi \sqrt{\frac{K_{y}}{M}}$), given that:(11)$$K0_{Q}=\frac{K_{xy\;}}{2M\omega_{0}}$$

We can do so for different values of ε, in order to determine the evolution of the standard deviation of $\left|K0_{Q}\right|$ versus ε. Figure 6 shows that really small variation of the bending beams width can generate high quadrature level (up to a RMS value of 1220 °/s for $\varepsilon /e_{0}=0.9\%$).

Then, we can compare our model to the measurements.

## 5. Electrical Measurements of the Amplitude of the Quadrature Signal

### 5.1. Principle

As we work with high Q-factor gyroscope $\left(Q>50\cdot 10^{3}\right)$ and low angular rate (vertical earth rotation), Equation (1) can be simplified as:(12)$$\left(\begin{array}{c} \ddot{x}\\ \ddot{y} \end{array}\right)+\left[\begin{array}{cc} \omega_{x}{}^{2} & \frac{K_{xy}}{M}\\ \frac{K_{xy}}{M} & \omega_{y}{}^{2} \end{array}\right]\left(\begin{array}{c} x\\ y \end{array}\right)=\left(\begin{array}{c} 0\\ 0 \end{array}\right)$$

If $\omega_{x}=\omega_{y}=\omega_{0}$, then the eigenmodes of oscillation, $\omega_{1}$ and $\omega_{2}$, can be expressed as follows:(13)$$\omega_{1}{}^{2}=\omega_{0}{}^{2}+\frac{K_{xy}}{M}\;\omega_{2}{}^{2}=\omega_{0}{}^{2}-\frac{K_{xy}}{M}$$

Furthermore, if $\sqrt{\frac{K_{xy}}{M}}\ll \omega_{0}$, then:(14)$$\frac{\left(\omega_{1}-\omega_{2}\right)}{2}\;\approx \;\frac{K_{xy\;}}{2M\omega_{0}}$$

Thus, according to Equations (11) and (14), we have:(15)$$\frac{\left(\omega_{1}-\omega_{2}\right)}{2}\;\approx \;K0_{Q}$$

Hence, if we are able to cancel the frequency mismatch (i.e., $\omega_{x}-\omega_{y}=0$), we can estimate the amplitude of the quadrature signal by measuring $\omega_{1}$ and $\omega_{2}$.

### 5.2. Measuring Bench

A dedicated bench, illustrated in Figure 7, has been developed to measure the quadrature error of each MEMS. The MEMS is mounted on a homemade interface board allowing us to retrieve the electrical contacts with the electrodes of the gyroscope. The resonator contains combs ensuring multiple functions such as the frequency mismatch compensation, sense and drive [16]. A PXIe-1078 bench produced by National Instruments, with multiple modules and a custom-built LabVIEW^TM^ V15 (schematized in Figure 8), was used to generate and record the excitation and detection signals. Each measurement was performed in three steps.

In the first step, we apply an appropriate DC voltage on the combs dedicated to the frequency mismatch compensation, in order to cancel the frequency difference between the *X* and *Y*-axis (i.e., $\omega_{x}-\omega_{y}\sim 0$). In the second step, a ring down test is performed: a DC voltage, supplied by a generator, and a white noise voltage signal, filtered in a band close to the resonance frequency of the axis *x* and *y*, is generated by the PXIe bench during two seconds (after these two seconds the excitation is switched off). Both of these voltages are then applied to the combs devoted to the drive function (both of these voltage pass through a custom-built filter to avoid any excitation noise). The PXI bench simultaneously records the decay of the resonator displacement via the sense combs (one set of combs for the *x* axis and another for the *y* axis) for a several seconds. Before being read by the PXI bench, the sense signals are amplified via a homemade charge amplifier, in order to convert the few nA of the transducer into dozens of mV. In the third step, our program filters the sense signals, in order to limit high and low frequency noise, and computes their power spectral density (PSD). Then, the program automatically find the maximum peaks of the PSDs and their frequencies, which are the $\omega_{1}$ and $\omega_{2}$ frequencies of Equation (15).

### 5.3. Experimental Results

#### 5.3.1. Precision of the Measure

In Section 5.1, we provided the Equations (13) and (15), which allowed us to estimate the quadrature bias, under the assumption that the frequency mismatch is fully compensated. But one can argue that the accuracy limits of the frequency mismatch compensation have an impact on the quadrature level calculation accuracy. So, if we now consider that $\omega_{x}\neq \omega_{y}$, the eigenmodes of vibration of the system (12), $\omega_{I}$ and $\omega_{\mathrm{II}}$, can be written as:(16)$$\omega_{I}{}^{2}=\frac{\omega_{x}{}^{2}+\omega_{y}{}^{2}+\sqrt{{\left(\omega_{x}{}^{2}-\omega_{y}{}^{2}\right)}^{2}+4{\left(\frac{K_{xy}}{M}\right)}^{2}}}{2}\;\omega_{\mathrm{II}}{}^{2}=\frac{\omega_{x}{}^{2}+\omega_{y}{}^{2}-\sqrt{{\left(\omega_{x}{}^{2}-\omega_{y}{}^{2}\right)}^{2}+4{\left(\frac{K_{xy}}{M}\right)}^{2}}}{2}$$

Thanks to Equation (13), it is possible to estimate the evolution of the error made in our measure, i.e., $\left(\frac{\left(\omega_{I}-\omega_{\mathrm{II}}\right)}{2}-\left|\frac{K_{xy\;}}{2M\omega_{0}}\right|\right)$, regarding the frequency mismatch $\left(\frac{\omega_{x}-\omega_{y}}{2\pi }\right)$.

As we can see in Figure 9, when $\omega_{x}=\omega_{y}$, the final frequency split $\frac{\left(\omega_{I}-\omega_{\mathrm{II}}\right)}{2}$ is exactly equal to the quadrature bias which is understandable, as we are in the exact same conditions as Section 5.1. Furthermore, for the same frequency mismatch, the smaller the value of $K_{xy}$, the higher the error made in the evaluation of $\left|K0_{Q}\right|$. This could represents a serious issue, as for high value of frequency mismatch (e.g., $\frac{\omega_{x}-\omega_{y}}{2\pi }=1\;\mathrm{Hz}$), the absolute value of the quadrature bias can be overestimated by 300%, for a real value of 47°/s.

Therefore, to ensure that the frequency mismatch is correctly compensated during the evaluation of $\left|K0_{Q}\right|$, the frequency mismatch tuning electrodes are used. By adjusting the voltage applied to these electrodes, we are able to find the ideal DC voltage where the final frequency split $\frac{\left(\omega_{I}-\omega_{\mathrm{II}}\right)}{2}$ is minimum. In order to validate this method, the experiment is carried out on the MEMS gyroscope number 1, as shown in Figure 10.

The absolute value of the final frequency split of the gyroscope (since one cannot tell which peak of the PSD corresponds to the $\omega_{1}$ peak) is reduced by increasing the voltage and reaches its minimal value when the voltage is 3.42 V. When the voltage continues to increase, the value of $\left|\frac{\left(\omega_{I}-\omega_{\mathrm{II}}\right)}{2}\right|$ increases. As a result, the absolute value the quadrature error of the gyroscope can be estimated with precision by applying the DC voltage on the tuning electrodes of frequency mismatch, minimizing the value of $\left|\frac{\left(\omega_{I}-\omega_{\mathrm{II}}\right)}{2}\right|$.

#### 5.3.2. Discussion of the Results

The absolute value of the quadrature error was measured on 24 MEMS with the design as the one simulated in parts 3 and 4. The results are shown in Figure 11. A RMS value of 1210°/s was calculated for the measurements of $\left|K0_{Q}\right|$, which is in the same order of magnitude as the results obtained through the numerical simulations (as mentioned in Section 4. That means that, for future gyroscope designs, it is necessary to consider the bending beams width variation in order to avoid too many MEMS with high quadrature error.

It is important to notice that, even if our model match (for $\varepsilon /e_{0}=0.9\%$) with the measurements, it nevertheless does not mean that the variation of the bending beams width is the only cause of quadrature error. Indeed, this error may have many origins such as the design of the resonator, broken springs, length or any other geometrical variation. However:The length variation effect on the quadrature bias remains tiny, as the length of the bending beams is large compared to their width, so a variation of the length has a smaller impact on the ratio ${\left(e/l\right)}^{3}$, which contributes to the stiffness of a bending beam [40], than a variation, of the same quantity, of the width;The value of $K_{xy\;}$ for our gyroscope with a ruptured spring, which its stiffness matrix corresponds to a zero matrix, is equal to several dozens of N/m. This would generate a much higher quadrature error (>15,000°/s) than the one we measured. Thus, we can say that we did not characterise a gyroscope with such a defect.

Hence, we can only say that the width variation of the bending beams represents the best explanation regarding the variation of the quadrature error in gyroscopes.

## 6. Conclusions

In this paper, we proposed a method to evaluate the amplitude of the quadrature signal caused by a variation of the bending beams width. We also have developed a dedicated bench to measure the quadrature error on the MEMS gyroscopes. It has been shown that such silicon etching defects can generate quadrature error up to a root mean square (RMS) value of 1220°/s for a bending beam width defect of 0.9%. Moreover, it turns out that this discrepancy is of the same order of magnitude as the one measured on the gyroscopes. As the consideration of the bending beams width defects is needed in order to avoid high quadrature error in MEMS gyroscopes, one can run the same kind of simulations as the ones presented with his own bending beams design.

The results of this study are particularly interesting since high quality factor ($Q>50\cdot 10^{3}$) resonators are required to improve the performance and the sensitivity of MEMS gyroscope [41]. To achieve such high Q, it is necessary to reduce thermoelastic damping, which is one of the major energy losses in micromechanical resonators [42]. But, to do so, it is necessary to reduce the width of the bending beams of the vibrating system. Yet, the variation of the bending beams width could then have a significant impact on the device performances (the ratio $\varepsilon /e_{0}$ would be increased). Thereby, we need to find a compromise between the quadrature error discrepancy and the enhancement of the Q-factor.

To overcome this issue, we can either reduce ε, i.e., the maximum variation of the bending beams width, or try to cancel the quadrature error. The first option requires to significantly improve the geometric accuracy of the etching technology, while the second one requires a specific architecture of the transducers and a control electronic [13,27]. However, doing so might make the nonlinear regime of the resonators easily reachable, which could decrease the sensor performances [43]. Another compromise needs therefore to be sought between the quadrature error cancelation and the nonlinear effects.

Further research is underway to investigate the real variation of the bending beams width via scanning electron microscopy (SEM) observations of our resonator.

## Figures and Tables

**Figure 1 micromachines-13-00655-f001:**
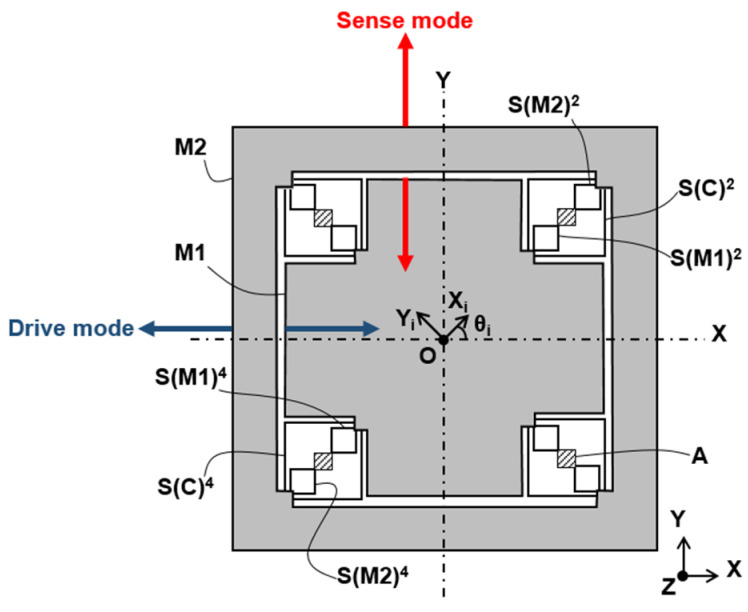
Conceptual view of the gyroscope [34].

**Figure 2 micromachines-13-00655-f002:**
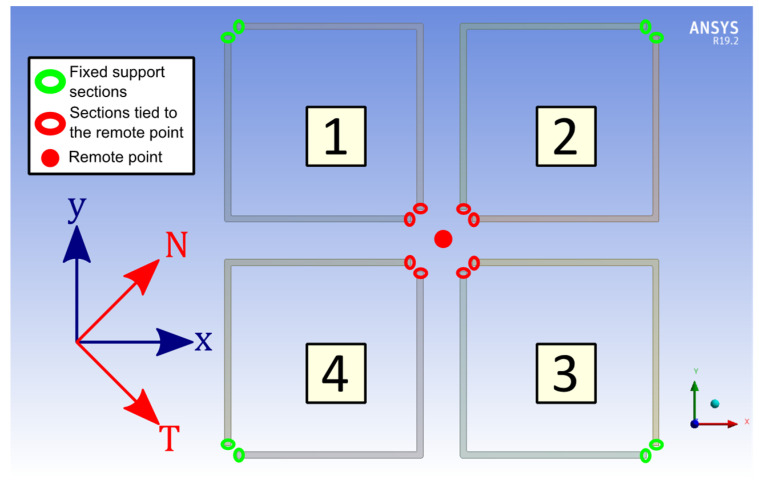
Model used during simulations.

**Figure 3 micromachines-13-00655-f003:**
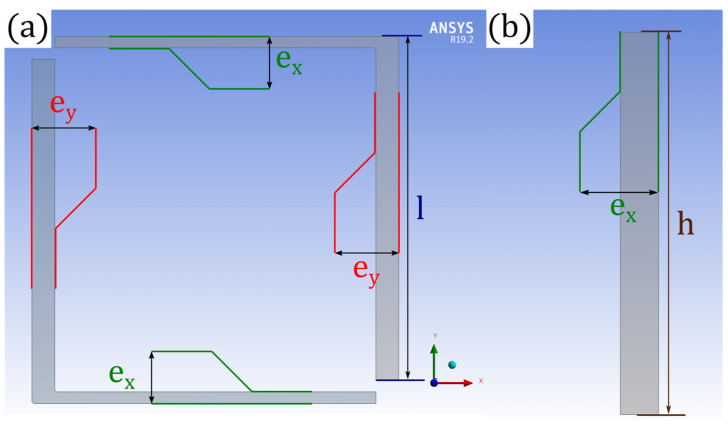
Representation of the geometrical parameters of the spring 1: (**a**) view from XY plan; (**b**) cross section of one beam viwed from ZY plan.

**Figure 4 micromachines-13-00655-f004:**
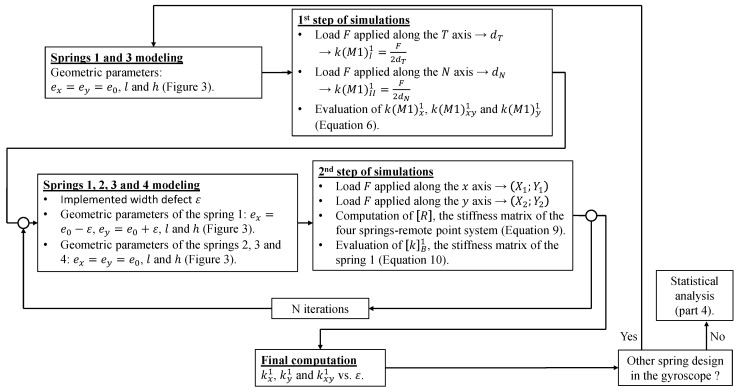
Synoptic diagram of the simulation procedure. Where ‘N iterations’ represents the number of iterations needed in order to run polynomial fits with a sufficient database for the values of ${\left[k\left(M1\right)\right]}_{B}^{1}$ terms.

**Figure 5 micromachines-13-00655-f005:**
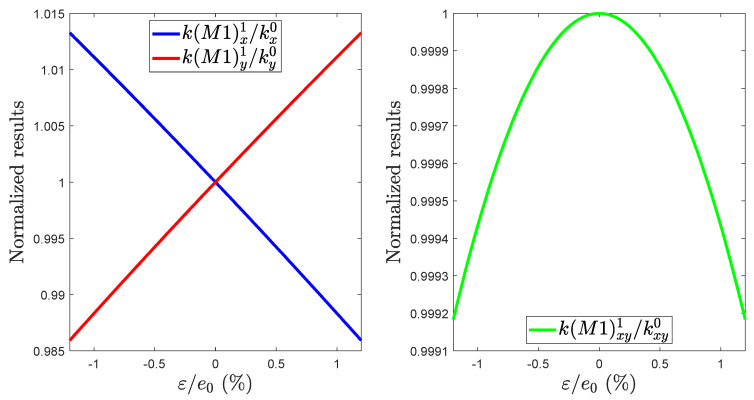
Normalized $k{\left(M1\right)}_{x}^{1}$, $k{\left(M1\right)}_{y}^{1}$ and $k{\left(M1\right)}_{xy}^{1}$ values as a function of the width variation.

**Figure 6 micromachines-13-00655-f006:**
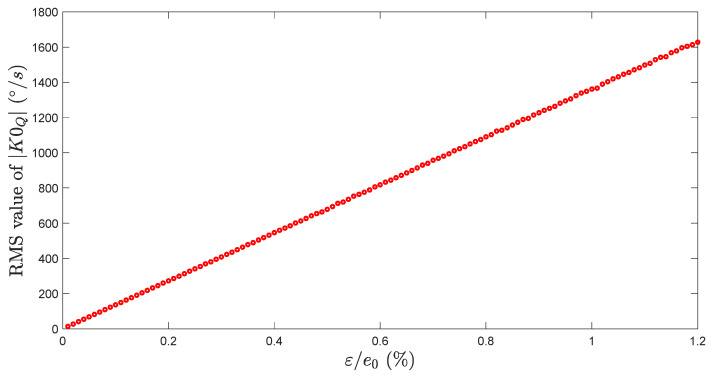
RMS value of $\left|K0_{Q}\right|$ vs. standard deviation of the beam width variation.

**Figure 7 micromachines-13-00655-f007:**
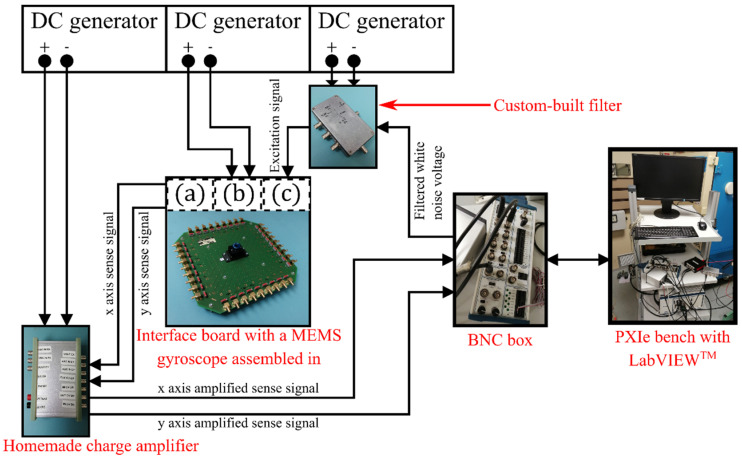
Scheme of the bench developed to measure the quadrature error: (a) contact with the *x* axis and *y* axis sense electrodes, (b) contact with the frequency mismatch compensation electrodes, and (c) contact with the *x* axis drive electrode(s).

**Figure 8 micromachines-13-00655-f008:**
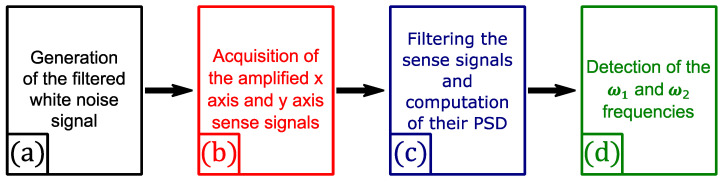
Schematic of the LabVIEW^TM^ program.

**Figure 9 micromachines-13-00655-f009:**
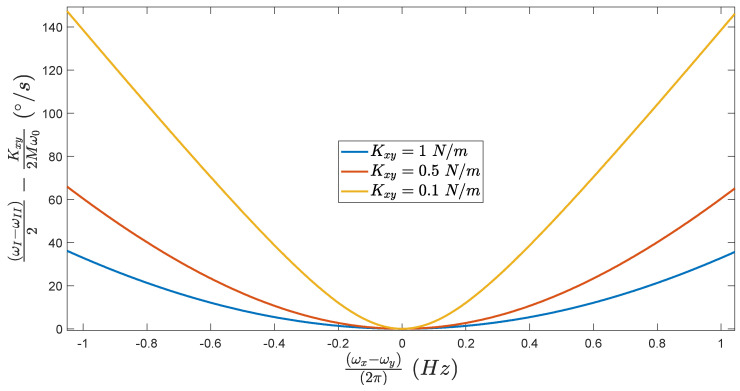
Evolution of the error made in our measure, regarding the frequency mismatch $\left(\frac{\omega_{x}-\omega_{y}}{2\pi }\right)$, for different value of coupling stiffness.

**Figure 10 micromachines-13-00655-f010:**
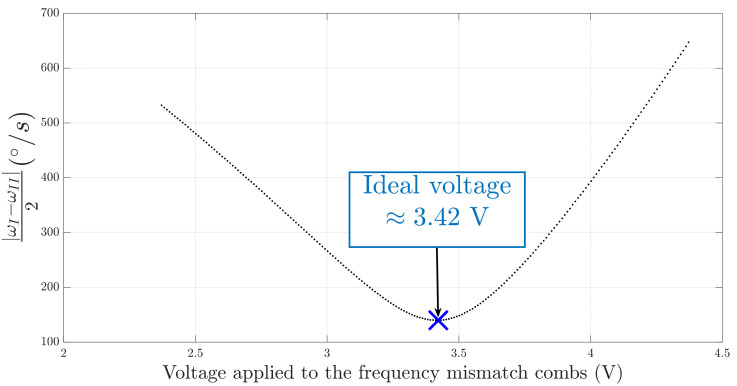
Frequency mismatch minimization experiment on gyroscope n°1.

**Figure 11 micromachines-13-00655-f011:**
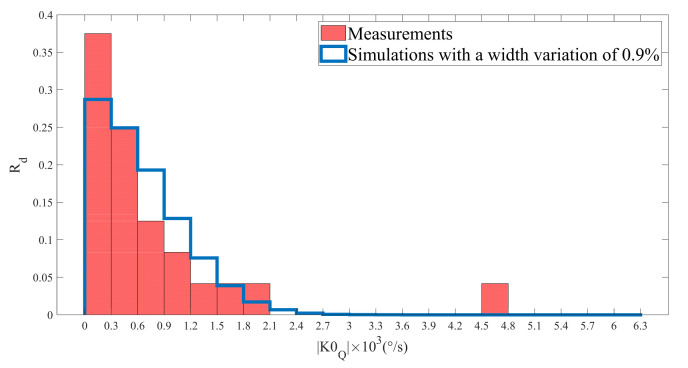
Comparison of the measurements of $\left|K0_{Q}\right|$ with the simulations. $R_{d}$ denotes the ratio of devices within a quadrature error interval to total number of devices.

## Data Availability

The data are not publicly available due to confidentiality purposes.

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
