# Peer review of "The Effect of the Bending Beam Width Variations on the Discrepancy of the Resulting Quadrature Errors in MEMS Gyroscopes"

_micromachines, 2022, doi:10.3390/mi13050655_

Round 1
Reviewer 1 Report
This paper presents a method to determine the evolution of the stiffness matrix of the gyroscope springs with respect to the inhomogeneity of the bending beams width of the springs through Finite Element Analysis (FEA). It is proved that the width defect of the bending beam needs to be considered to avoid high orthogonality error.I recommend rejection. The paper needs to rewritten before being considered again. Some of my comments are
- The overall structure of the article is not very logical. Please re frame itï¼›
- The title of the article is "the effect of the bending beam width inhomogenities on the diversity of the resulting square errors", but the description of "beam width variation" does not correspond to the title.
Author Response
We are very much thankful to the reviewer. We have revised our present research paper in the light of the reviewer useful suggestions and comments. We hope our revision has improved the paper to the level of the reviewer satisfaction and the required quality for the journal. Answers to comments/suggestions/queries are as follows.
All the modifications made are in red in the revised version of the manuscript.
- The paper was completely revised and the authors took into account all the comments of the reviewers.
- Indeed, the word "inhomogeneities" is inadequate and it is replaced by "variations" in the title and throughout the text.
Reviewer 2 Report
Thanks for your paper. I have the following comments:
- Your introduction is fine, but recently gyroscopes with bias stability under 0.1deg/hr are reported in the literature from universities and companies (see CRH03 from Silicon Sensing). Please give citation to these works.
- Most of your references are not properly cited. Please provide the conference/journal name, other than just saying IEEE.
- Please double check your paper. Try to avoid simple mistakes as page 3, line 118: "gyroscope springs. , regarding the spring..."
- For Equation (1) and martix-C please clearly state your assumption for angular gain. It seems that you take it equal to 1 (ideal and maximum value).
- I understand the confidentiality issue, but please explain your Figure 1 little bit more. Identify the masses as the drive and sense modes. You can also talk about desired degree of freedoms for M1 and M2.
- Your simulation procedure can be simplified. A diagram would really help. Detailed explanation is fine, but please provide a simpler explanation.
- Your experimental method seems unorthodox for a functional gyroscope. Why don't you simply run your device as a gyroscope and characterize the quadrature error? Please comment on it. I am not opposing your method, but it seems that it is sensitive to errors.
- Your Figure 10 is not very clear for me. What is the y-axis? What does "proportion of gyroscope" mean? Is it the ratio of devices of certain quadrature error bin to total number of devices? If so, why doesn't it add up to 100%?
Author Response
We are very much thankful to the reviewer for the deep and thorough review. We have revised our present research paper in the light of the reviewer useful suggestions and comments. We hope our revision has improved the paper to the level of the reviewer satisfaction and the required quality for the journal. Answers to comments/suggestions/queries are as follows.
All the modifications made are in red in the revised version of the manuscript.
- Citations to some of these works have been added (citation [17] and [19]).
- Indeed, most of the references were not properly cited. The ‘References’ section is updated.
- The cited mistake (page 3 line 118) is corrected, and the manuscript was carefully proofread.
- [C] represents the Coriolis matrix and the angular velocity around the z-axis. If we clearly understood your remark, no assumption needs to be added in this design, as there is no angular gain in this device, which does not contain any amplification mechansim.
- Figure 1 is updated, so one can see the drive and sense modes. A sentence is added (in red) in page 3 in this regard.
- As a matter of fact, the simulation procedure (and its explanation) is quite complex. We then added the figure 4 (a diagram of the simulation procedure) and a sentence (in red, page 6).
- As you point out, the classical way to characterize the quadrature error is to run the MEMS as a gyroscope. Even though this ‘classic’ method allows one to make accurate measurements of the quadrature error, it usually required a MEMS with on a control electronic and some fine-tuning, which is time consuming.
With the method presented in the paper, one only need to put the MEMS on the interface board and press a button in order to perform the measurement. Furthermore, with this method, the measurement is automated and usually lasts 2min for each MEMS. This then allows the characterization of the quadrature error on an industrial level and reduce the production cost (the MEMS with a high quadrature bias will not be mounted as a gyroscope). This method is indeed sensitive to errors, but we proved that we are able to tune the resonator in such a way that the error made on the measurement is negligible (moreover, we don’t talk about that in this article, but with this method, we measure quadrature errors close to the values that we obtained when we run the device as a gyroscope). - The term ‘proportion of gyroscope’ is indeed inadequate. So, we changed it, as you suggested, with ‘Ratio of devices within a quadrature interval to total number of devices’. It didn’t add up to 100% because one bin was missing, so we changed the figure to make it visible.
Reviewer 3 Report
The paper investigates the effect of beam width inhomogeneity on the quadrature errors in gyroscopes. The simulation results are in good agreement with the experimental results, which show the correctness of the analysis. However, the practicality of the conclusions is questioned. The manuscript needs a major revision. My detailed comments are as follows.
- The author should carefully check the format and the coherence of the manuscript. For example:
On page 3, line 118, two punctuation marks ‘.,’ appear in succession;
On page 7, line 209, which figure are the results?
In the same line, ‘and the’ is not in Times New Roman font;
On page 8, line 242, the variable ‘m’ should be ‘M’;
On page 9, Figure 7(c), ‘computaton’ should be ‘computation.’
- On page 7, line 222, the author should explain why the normal law’s variance is set to ε/3.
- In Section 3, only L-shape beams are investigated. Can this simulation apply to other kinds of beams to improve their generality?
- After we know the effect of beam width inhomogeneity, how can we apply the conclusions to suppress the quadrature output of gyroscopes? The practicability and application significance of this paper is in doubt.
Author Response
We are very much thankful to the reviewer for the deep and thorough review. We have revised our present research paper in the light of the reviewer useful suggestions and comments. We hope our revision has improved the paper to the level of the reviewer satisfaction and the required quality for the journal. Answers to comments/suggestions/queries are as follows.
All the modifications made are in red in the revised version of the manuscript.
- All the mistakes you pointed out are corrected (visible in red in the document) and the manuscript was carefully proofread.
- The choice of the normal law variance is qualitative. The idea is that, with this value, 99,7% of the bending beams have a width comprise between e0-ε and e0+ε. One can take any value for the variance, if it suits better for one’s fabrication tolerances. A sentence is added (in red) page 8 to explain this choice
- One paragraph is added (in red) page 7, in order to give an idea of the suitability of this type of simulations to other kinds of spring/beam design.
- The aim of this article is not to give to the reader a method to suppress the quadrature error. Here the idea is to show and prove the effect of the beam width variations on the quadrature dispersion. In the end, one needs to consider these variations during the design of the gyroscope. Then, one can decide if its manufacturing process needs to be improved, if the control electronic of its gyroscope needs to be enhanced or if the design needs to be changed because it’s very sensitive to fabrication variations. A paragraph is added (in red) to the ‘Conclusions’ section in regard to your comment.
Round 2
Reviewer 1 Report
This paper proposed a method to evaluate the amplitude of the quadrature signal caused by a variation of the bending beams width. We also have developed a dedicated bench to measure the quadrature error on the MEMS gyroscopes. And I have some other suggestions that may help the paper to be better for readers:
1. The words "Part 0" and "section 0" still appear in section 5.3, which should be changed .
2. The English should be further polished, some sentences should be written in a better format.
Author Response
We are very much thankful to the reviewer. The paper was revised and carefully proofread while taking into account all the comments of the reviewers.
Reviewer 2 Report
Thank you for the changes. Although the quality of the paper is significantly improved, the block diagram and in general the explanation of your approach can be improved.
- In Eq. (10) it is not very clear how you extract the individual spring terms 2, 3, and 4. At the end of page 7, you say that "first step could be used". Is there an alternative way of doing it?
- In your Figure 4, what are the iterations for? It would be great if you explain it explicitly such as "Iterations for..."
Author Response
We are very much thankful to the reviewer. We have revised our present research paper in the light of the reviewer useful suggestions and comments. We hope our revision has improved the paper to the level of the reviewer satisfaction and the required quality for the journal. Answers to comments/suggestions/queries are as follows.
All the modifications made are in red in the revised version of the manuscript.
-
We added a sentence (page 7, line 201-203) in order to clarify our method to extract the individual spring terms 2, 3, and 4. Regarding the end of page 7, to our knowledge, this is the only efficient way to do so. This may open the way for researchers to propose other alternative methods.
- We added two sentences (page 6, figure 4; page 7, lines 210, 213 and 214), to explain that the iterations concern the construction of a database for polynomial fits.
Reviewer 3 Report
Thank you very much for your revision. I have no specific questions about this version.
Author Response
We are very much thankful to the reviewer.